# [Re] Benchmarking LLM Capabilities in Negotiation through Scoreable Games

**Jorge Carrasco Pollo** [†]                                    *jorge.carrasco.pollo@student.uva.nl*
*Informatics Institute*
*University of Amsterdam*

**Ioannis Kapetangeorgis** [†]                              *ioannis.kapetangeorgis@student.uva.nl*
*Informatics Institute*
*University of Amsterdam*

**Joshua Rosenthal** [†]                                     *joshua.rosenthal@student.uva.nl*
*Informatics Institute*
*University of Amsterdam*

**John Hua Yao** [†]                                              *john.yao@student.uva.nl*
*Informatics Institute*
*University of Amsterdam*

## Abstract

Large Language Models (LLMs) demonstrate significant potential in multi-agent negotiation tasks, yet evaluation in this domain remains challenging due to a lack of robust and generalizable benchmarks. Abdelnabi et al. (2024) introduce a negotiation benchmark based on Scoreable Games, with the aim of developing a highly complex and realistic evaluation framework for LLMs. Our work investigates the reproducibility of claims in their benchmark, and provides a deeper understanding of its usability and generalizability. We replicate the original experiments on additional models, and introduce additional metrics to verify negotiation quality and evenness of evaluation. Our findings reveal that while the benchmark is indeed complex, model comparison is ambiguous, raising questions about its objectivity. Furthermore, we identify limitations in the experimental setup, particularly in information leakage detection and thoroughness of the ablation study. By examining and analyzing the behavior of a wider range of models on an extended version of the benchmark, we reveal insights that provide additional context to potential users. Our results highlight the importance of context in model-comparative evaluations.

## 1 Introduction

The growing use of LLMs in multiagent systems has spurred demand for robust evaluation benchmarks. While existing benchmarks assess capabilities such as reasoning and task completion (Polo et al., 2024; Wang et al., 2024; White et al., 2024), their ability to measure nuanced social interactions, such as negotiation and consensus-building, remains understudied. Brookins and DeBacker (2023) and Akata et al. (2023) investigate LLM behavior in structured negotiation games, however they primarily make use of simple prisoner's dilemma scenarios that necessitate greater effort to generalize to complex real world negotiations. As these benchmarks are also static, their relevance is also called into question as increasingly powerful LLMs are released. For this reason, Davidson et al. (2024) employs a dynamic benchmark with adjustable complexity. However, their scope is limited to the case of only two negotiating agents.

In response, Abdelnabi et al. (2024) introduce a benchmark[‡] grounded in Scoreable Games (Susskind, 1985), a round-based multi-agent discussion format designed to teach negotiation. This proposed framework evaluates

---

[†]Equal contribution.

[‡]Original implementation: https://github.com/S-Abdelnabi/LLM-Deliberation

the ability of LLMs to reach agreements through role-based dialogue, which offers a promising tool for assessing cooperation in real world multi-agent settings. However, the usability of such benchmarks hinges on how well the experiments generalize, and how fair the outcomes are to each model. More specifically, usability depends on how wide the range of models the benchmark is capable of evaluating, and whether a user can derive consistent comparisons between models using the benchmark. There are a multitude of articles that highlight the gaps in LLM benchmarking in general, including the risk of leakage (Xu et al., 2024) and inconsistent metric rigor (Sun et al., 2024), which can invalidate benchmark results and create unfair comparison models. However, few studies critically evaluate negotiation-focused benchmarks such as the one proposed by Abdelnabi et al. (2024).

In this paper, we reproduce and extend the work of Abdelnabi et al. (2024)[§]. Our contributions are:

- A reproduction study analyzing the ability of the benchmark to generalize across a wider variety of untested models, exposing and addressing gaps in weaker model evaluation and replicating adversarial LLM behaviors;

- Identification of unfair comparisons in the original experimental setup due to inconsistent model performance across benchmark dimensions and sensitivity to ablation configurations;

- Finding and fixing several technical limitations in the code;

- Enhanced evaluation transparency via supplementary scoring metrics;

- Demonstration of the limited adaptability of the framework through a lack of game diversity.

## 2   Scope of reproducibility

The benchmark proposed by Abdelnabi et al. (2024) aims to study the capabilities of LLMs to cooperate and negotiate through a round table format of competition, where agents are instructed to keep their preferences hidden. Moreover, through altering prompts, score functions, and behavioral characteristics, the authors claim the benchmark can be easily adaptable to suit the testing requirements of new models. As an open source benchmark, the original work provides further transparency on the capabilities of language models to negotiate through empirical evaluation.

Drawing on frameworks for benchmark evaluation (Reuel et al., 2024; Xu et al., 2024), we operationalize concerns of generalizability and fairness into testable claims and validate them through experiments across several models and negotiation scenarios.

**Claims**

We aim to address a series of claims, both implicit and explicit, that we identify in the original paper. The claims are as follows:

- Using the framework provided, a user can objectively compare skills in multi-agent negotiation between different LLMs. This claim will be referred to as the **benchmark claim**. We demonstrate that, due to inconsistencies in model performance across games, this claim is unsupported by evidence.

- This framework provides a collection of games that are diverse and easily adjustable. This will be referred to as the **adjustments claim**. We show that while games are easily adjustable, there is insufficient evidence of diversity in both the original provided games and newly generated games using built-in methods.

- When prompted to, language models exhibit greedy and adversarial behavior, negatively affecting group outcomes. This will be referred to as the **behavioral claim**. We empirically verify that this claim is true through reproduction.

We begin by verifying these claims experimentally, followed by a discussion of their validity. We focus primarily on the **benchmark claim**, as this is put forward as the primary contribution of the original paper.

---

[§]Our implementation: https://github.com/joshrosie/FACT29

## 3    Methodology

We begin by formalizing the negotiation games as described by Abdelnabi et al. (2024), including all game variants and evaluation metrics from the original work. To reproduce the experiments within our computational constraints, we select smaller models such as GPT-4o mini and implement quantization tooling for larger open-source models. We then document critical modifications to the original open-source codebase, such as refactoring the leakage detection logic. Finally, we design experiments targeting the original claims, including extensive ablation studies, cross-game model performance comparisons, and an in-depth analysis of game adjustability.

### 3.1    Implementation Details

#### 3.1.1    Game Description

As defined in the original paper, each game consists of $N$ distinct parties $\{p_i\}_{i=1}^N$, where $N = 6$ by default, who negotiate over five predefined issues (e.g., building a tower). Each issue has a different number of possible sub-issue values, which represent the different possible options for an issue (e.g., making the tower 3, 4 or 5 stories tall). These issues represent various aspects of the negotiation scenario, requiring agents to deliberate and reach a consensus that satisfies their individual and collective objectives.

The negotiation begins with a predefined deal from $p_1$ (the original paper provides a default deal for each game). Afterward, players propose deals in a randomized sequence for $R$ rounds ($R = 24$ by default), and after the last round, $p_1$ proposes a final deal. Since the deals are proposed sequentially, we use $\pi^{(t)}$ to denote a deal proposed in the $t - th$ round, and we denote the final deal as $\pi^{R+1}$.

Each agent has an associated utility function $U_{p_i}(\cdot) : \Pi \to \mathbb{N}$ and a threshold $\tau_{p_i}$. A player can reach a maximum utility of 100 for a deal, and hence the threshold is a number between 0 and 100. A deal is said to be acceptable if its utility exceeds the threshold for at least $N - 1$ players (5 in the case of 6 players). Furthermore, the two players with veto power, $p_1$ and $p_2$, must also accept the deal.

#### 3.1.2    Game Variants

From Abdelnabi et al. (2024) we identify three primary attributes to game settings that serve to alter negotiation dynamics:

**Setting Updates:**    A language model is prompted to generate the text for the agent prompts, which introduces the motivations of the different agents.

**Score Function Updates:**    All utility functions $U_{p_i}(\cdot)$, as well as the minimum acceptable thresholds $\tau_{p_i}$, are manually curated by the original authors.

With these parameters, 5 pre-configured games are provided by the authors of the original paper: The **base** and **base rewritten** games share the score functions but have different setting descriptions. **Games 1**, **2**, and **3** have settings and score functions.

**Agent Behavior Updates**    Another way the games change is through agent prompting. By default, agents are instructed to be cooperative and to prioritize achieving a deal. However, with the goal of evaluating the ability of models to adopt prompted behavior, agents can be prompted to assume different strategies: a greedy approach, where they prioritize maximizing their personal score, or an adversarial approach, where they actively seek to prevent a deal from being reached.

#### 3.1.3    Game Evaluation

We evaluate the dynamics of the negotiation process of the agents using the same metrics as Abdelnabi et al. (2024) as well as additional metrics that we proposed, which are discussed in Section 3.2.1. We use the percentage of deals achieved over 20 game iterations as the primary evaluation metric. These metrics from the original paper are:

**5/6-way:**    Acceptable deals proposed in the final round (i.e. whether $\pi^{(R+1)} \in \Pi_{acc}$), where

$$\Pi_{acc} = \{\pi \in \Pi | U_{p_1}(\pi) \geq \tau_{p_1}, U_{p_2}(\pi) \geq \tau_{p_2} \text{ and } |\{p_i \in P | U_{p_i}(\pi) < \tau_{p_i}\}| \leq 1\}.$$

**6-way:** Deals acceptable by 6 players in the final round (i.e. whether $\pi^{(R+1)} \in \Pi_{hard}$), given by

$$\Pi_{hard} = \{\pi \in \Pi | \forall p_i \in P, U_{p_i}(\pi) \geq \tau_{p_i}\}.$$

**Any:** Whether at least one of the deals proposed by $p_1$ belongs to $\Pi_{acc}$.

**Leakage:** How often an agent discloses its private scoring information despite being instructed not to. In the original paper, leakage is measured using a GPT-4 model that receives the entire conversation and judges whether leakage occurred. We note that most leakage was induced by bugs, discussed in Section 3.1.5.

**Wrong deals:** Proportion of deals proposed where the score of the proposing agent falls below its threshold.

### 3.1.4 Model Descriptions

We extend the original study by evaluating the benchmark on a broader range of models (including smaller and open-source architectures) to test whether the results of the benchmark remain robust beyond the originally tested systems.

All models are sourced from Hugging Face[*] and used in their pre-trained form without fine-tuning. Due to computational constraints, we quantize larger models using *bitsandbytes*[†] to fit within available resources (see Appendix A, Table 7).

For subsequent experiments, we filter models to only those demonstrating no leakage under the evaluation framework of Abdelnabi et al. (2024). This ensures fair comparisons, as leakage corrections (discussed in Section 3.1.5) could disproportionately affect results across models. To further assess generalizability, we also evaluate GPT-4o mini[‡] and GPT-4o[§], testing whether claims hold for closed-source architectures.

### 3.1.5 Code Adjustments

**Addressing Leakage:** The original paper uses GPT-4 to process whether a score leakage has occurred in a given interaction. However, an inspection of the original implementation reveals that the reported leakage metrics are heavily confounded by structural code behavior, as opposed to purposeful goal leakage by the agents to obtain an advantage. When models deviate from formatting rules (e.g., omitting `<ANSWER>` tags), the system automatically exposes confidential outputs. This pattern was observed to comprise the bulk of reported leakage events in our runs. It remains unclear whether this is a subversive attempt by the model to gain negotiation advantage, however by separating this particular case we obtain a more interpretable and precise definition of leakage.

We implement this by handling malformed public responses containing illegal keywords or missing the required structural markers. We weaken the requirements on public answer extraction, such that responses with minor formatting issues are also processed into a public answer. In the case where no valid public answer can be extracted, we terminate negotiations, exclude them from metric calculations, and log the run as a failure. This approach separately categorizes outputs containing a few commonly seen formatting errors, giving a more interpretable leakage metric. These results can be found in Appendix D.

To quantify the remaining leakage, we flag instances where an answer contains illegal keywords such as `<plan>` or `<scratchpad>`.While the previous paragraph discusses handling malformed responses, this method explicitly focuses on measuring leakage through structured detection. In the original study, GPT-4 naturally adhered to formatting rules, avoiding leakage. However, smaller models often failed to do so, causing unintended leakage due to parsing errors. Our fix addresses this issue, enabling fair evaluation of these weaker models by ensuring their negotiation attempts are not discarded due to formatting inconsistencies. We validated our approach on GPT-4o mini, Qwen2.5-72B, and Mistral-Small, which followed the expected format and showed no leakage.

---

[*]https://huggingface.co/

[†]https://huggingface.co/docs/bitsandbytes/main/en/index

[‡]https://openai.com/index/gpt-4o-mini-advancing-cost-efficient-intelligence/

[§]https://openai.com/index/hello-gpt-4o/

While leakage remains a concern in some other architectures, our refined approach substantially mitigates its impact, ensuring that low-performing models can still be meaningfully evaluated without being unfairly penalized for formatting inconsistencies.

**Other Code Changes:** The criteria for deal acceptance are inconsistent. Firstly, the text specifies that a deal must "exceed" thresholds (implying strict inequality $>$), but the original implementation permits values greater than or equal to thresholds ($\geq$), as seen in the `check_agreement` function. To align with their implementation, we adopt $\geq$ in our replication. Secondly, we find that a condition within the provided evaluation script implements an unconditional relaxation of the threshold of $p_1$ by 10 units. In the prompts implemented in the original paper, it is stated that $p_1$ can get a 10 points bonus in case of unanimity, which is not consistent with their implemented evaluation. This game variation is not introduced in the original paper aside from a mention in Appendix F, which is only meant to apply to greedy games. As this can result in inflated performance metrics by increasing the size of the feasibility set, we do not adjust the threshold of $p_1$.

Additionally, we identify and correct a bug in the original evaluation, where when $p_1$ fails to propose a valid deal in the final round, the last valid deal from $p_1$ is by default considered as the final deal. A valid deal is thus accepted as one that can be successfully parsed, instead of a deal that while incorrectly formatted, still functions as one. This oversight leads to inflated *Final* metrics, as non-final deals are mistakenly used in the evaluation.

An overview of the code used to run experiments, along with a description of the changes made to improve the user experience of running multiple experiments can be found in Appendix A.1.1.

## 3.2 Experiments

We now describe the experiments designed to assess each claim as defined in Section 2.

### 3.2.1 Benchmark claim

The **benchmark claim** states that a user may use the proposed experiments as a tool to measure and compare the ability of language models to cooperate and negotiate. We assert that, for this claim to be true, these experiments must provide an interpretable comparison between models. Otherwise, it would be difficult for a user to draw conclusions about how the ability of a model to negotiate compares to another. For this purpose, the following experiments aim to both provide a more complete view of each step in the model comparison process, and of the games themselves.

**Base Game Performance:** We attempt to reproduce the findings on the **base game**. We will refer to this experiment as **Experiment 1**. With this experiment, we investigate the generalizability of this benchmark to further models.

**Ablations:** In the original paper, the ablation study was unclear as to the exact prompt adjustments made as well as why only a select portion of possible configurations were attempted. We reproduce the ablations performed in the original study as well as additional experiments to cover the full range of possible ablations. We will refer to this experiment as **Experiment 2**. We run the ablation study on more than one model, namely GPT-4o mini and Qwen2.5-72B. This is done to investigate whether an optimal ablation configuration for one model is optimal for others. As the ablated prompt templates are not provided by the authors, we create our own which can be found in Appendix B. We detail our full ablation configurations in Table 2.

**Extra Game Performance:** The original work repeats runs for **base, base rewritten, game 1, game 2,** and **game 3** to demonstrate a wider variety of performances across different games. However, there are no results for how models other than GPT4 perform on these different games. We evaluate three models — Mistral-Small, Qwen2.5-72B, and GPT-4o mini — across all negotiation games. We selected Mistral-Small and Qwen2.5-72B in accordance with the criteria laid out in Section 3.1.4, while GPT-4o mini tests generalizability to closed-source architectures. This analysis, labeled **Experiment 3**, compares negotiation performance across models to assess consistency and robustness.

**New evaluation metrics:** In the original paper, each game reported metrics based on the number of deals accepted by five or six parties. While this approach captures broad acceptability, several frameworks

exist to evaluate the success of automated negotiations (Endriss et al., 2006; Sanchez-Anguix et al., 2021; Tang and Ito, 2018). Building on these efforts, we analyze negotiation dynamics through the lens of social welfare—a framework particularly suited to assessing fairness and efficiency in multi-agent systems. To this end, we examine three social welfare metrics: (1) Utilitarian Social Welfare (USW), which sums the utilities of all players in a deal, this metric is equivalent to the performance reported by the original paper; (2) Egalitarian Social Welfare (ESW), which considers the minimum utility among players as a measure of fairness, this targets whether the interests of a single agent are being ignored; and (3) Nash Social Welfare (NSW) (Kaneko and Nakamura, 1979), which multiplies individual utilities to balance efficiency and equity, observing if utility is evenly spread across agents. These metrics define the foundation of **Experiment 4**, enabling a structured comparison of negotiation outcomes across different fairness paradigms. Importantly, these additional metrics are not intended as an expansion of the original benchmark; rather, they serve as a safeguard to verify that negotiations remain fair and to ensure that models do not implicitly collaborate against a single player, artificially inflating overall scores at the individual's expense.

**Baseline Models:** The original work proposes a repeated rule-based baseline in which each agent, in turn, modifies a previously proposed deal by incrementally optimizing the sub-options from highest to lowest priority until its minimum threshold is reached (or no further improvements are possible). However, because the implementation for this is not available and the notion of "importance" or "priority" is insufficiently defined, producing outcomes consistent with those reported in the original paper using an intuitive notion of priority is challenging. Consequently, we introduce an easy to reproduce baseline in which issues are selected in a random sequence rather than sorted by priority, preserving all other elements of the authors' procedure exactly as described. We refer to this as **Experiment 5**.

### 3.2.2 Adjustments claim

This claim affirms that the provided games are diverse and easily adjustable. In Section 3.1.2 we categorize the three main attributes that differentiate games: setting, score function, and behavior. As the impact of behavioral differences is discussed while analyzing the **behavioral claim** (Section 4.3), we investigate whether the games differ greatly in the setting and score function categories and how adjustable the given games are in said categories.

Regarding settings, the authors provide a prompt to generate new negotiation scenarios. After looking at the five existing games, we found that the setting in every game is based on a construction project. As there exist many other plausible negotiation settings, we investigate if alternative backgrounds are achievable with their provided prompt. To do this, we query the prompt ten times using the GPT-4 model to remain consistent with the methodology suggested by the original work. Finally, we construct an alternative version of the prompt with words such as "project" and "resources" removed to show that non-construction proposals are obtainable through the same method.

As for score function updates, the original work proposes to adjust games through (1) modifying player-specific minimum thresholds, (2) changing the number of players and (3) changes in the sparsity of the score function. While the sparsity of the scoring function is noted as a potential factor, it is never clearly defined. If sparsity is interpreted as the number of 0 valued weights in the score functions, it is not easily tunable, as a change in sparsity needs to keep the same semantics inherent in the preferences of each agent. For example, the relative orderings of the scoring functions of each agent should be maintained, and trivially changing values to 0 to increase sparsity could contradict this.

Sparsity being one factor, there is no straightforward way to quantify diversity in score functions. We approach this problem in **Experiment 6**, where we introduce candidate statistics:

- **Sparsity**: Counts the number of zero-valued sub-options throughout all score functions, reflecting how many issues are "ignored".

- **Intersection over Union(IoU)**: Measures average pairwise overlap among agent score functions, indicating similarity in preferences. See Appendix F for implementation details.

- **Deal Space**: Tracks the size of the set of all acceptable deals ($\Pi_{acc}$) and set of 6-way acceptable deals ($\Pi_{hard}$).

In Abdelnabi et al. (2024), the authors hypothesize that differences in sparsity explain the varying model performance on different games. By defining two measures (sparsity and IoU), we investigate quantitatively if this is valid. Deal space is investigated in the original paper, and we reproduce this calculation to assist in game comparison.

The original study briefly explores varying thresholds and player count. In our work, we test these hypotheses and expand the experiment on thresholds to a wider range. We refer to these as **Experiment 7**.

### 3.2.3   Behavioral claim

The original work investigates whether agents exhibit adversarial behavior when prompted, as well as how greed affects utility distribution. Their results show that in the greedy setting, the greedy player maintains a higher utility compared to the cooperative case, while in the adversarial setting, a targeted agent experiences lower utility than when not being targeted. We quantify these effects using GPT-4o mini and Qwen2.5-72B, as shown in Table 6 and Figure 3, and confirm that these trends persist across models. This replication, which we term **Experiment 8**, extends the generalizability of the original findings and provides a comparative evaluation of model responses to strategic behavior prompts.

## 4   Results

### 4.1   Benchmark claim

The results for **Experiment 1** are detailed in Table 1, which shows the previously defined performance metrics. The tested **base game** is very close to being solved by the DeepSeek models which, even at a quantized level, achieves near 100% performance in 5- and 6-way agreement. This aligns with the Abdelnabi et al. (2024)'s prediction that better performing models, improving with every new release, necessitate a benchmark capable of scaling in difficulty. 6-way agreement still remains a difficult goal to perfectly complete, but significant gains are also visible there.

Additionally, with our knowledge of the leakage bug and our subsequent fixes detailed in the methodology, we note a vastly increased variance in the **Leaked** column compared to the original paper. This can be explained by the inclusion of smaller models whose responses became parse-able with our changes.

| | Model | Final ↑ | | Any ↑ | Wrong ↓ | Leaked ↓ |
|---|---|---|---|---|---|---|
| | | 5-way | 6-way | | | |
| (4om) | GPT-4o mini | 55 | 5 | 90 | 2.69 | 0.58 |
| (gpt, 2024) | GPT-4o | 75 | 10 | 85 | 0.58 | 0.00 |
| (Touvron et al., 2023) | **Llama-2-13b (int8)** | 30 | 0 | 75 | 17.92 | 9.23 |
| (Grattafiori et al., 2024) | Llama-3-8B | 25 | 0 | 70 | 8.14 | 69.42 |
| (Grattafiori et al., 2024) | **Llama-3.3-70B (int4)** | 60 | 0 | 100 | 0.96 | 0.00 |
| (Qwen et al., 2025) | Qwen2.5-7B | 65 | 25 | 100 | 11.15 | 44.62 |
| (Qwen et al., 2025) | Qwen2.5-72B (int4) | 85 | 0 | 95 | 2.12 | 0.00 |
| (Abdin et al., 2024a) | Phi-3.5-mini | 10 | 10 | 50 | 12.87 | 40.77 |
| (Abdin et al., 2024b) | Phi-4 (int8) | 25 | 5 | 70 | 0.77 | 0.00 |
| (AI) | Ministral-8B | 25 | 0 | 50 | 12.88 | 7.12 |
| (mis) | Mistral-Small (int8) | 80 | 0 | 100 | 11.15 | 0.00 |
| (Jiang et al., 2024) | **Mixtral-8x7B (int4)** | 30 | 5 | 55 | 23.64 | 24.81 |
| (DeepSeek-AI et al., 2025) | DeepSeek-R1-Distill-Qwen-32B (int8) | 95 | 55 | 95 | 2.50 | 0.00 |
| (DeepSeek-AI et al., 2025) | DeepSeek-R1-Distill-Llama-70B (int4) | 90 | 65 | 90 | 0.19 | 0.00 |
| (OpenAI et al., 2024) | GPT-4 * | 81 | 33 | 100 | 1.40 | 0 |
| (Brown et al., 2020) | GPT-3.5 * | 20 | 8 | 33 | 19 | 25 |
| (Touvron et al., 2023) | Llama2-13b * | 57 | 10 | 82 | 16 | 14 |
| (Touvron et al., 2023) | Llama2-70b * | 76 | 19 | 95 | 11 | 22 |
| (Grattafiori et al., 2024) | Llama3-70b * | 60 | 21 | 100 | 4 | 2 |
| (Team et al., 2024) | Gemini Pro * | 45 | 0 | 70 | 13 | 6 |
| (Jiang et al., 2024) | Mixtral 8x7B * | 65 | 17 | 95 | 11 | 12 |

Table 1: Performance of models on the **base game** measured in %. Bolded model names indicate common models, quantized, with the original study. The results of models suffixed by "*" are taken from the original study (Abdelnabi et al., 2024).

In other words, Table 1 reproduces the findings of the original study using different models, showing plausible performances. Additionally we may observe that in spite of necessary quantization procedures, large models outperform their smaller, non-quantized counterparts in negotiation despite similar sizes in memory.

| Ablations (✓= No Ablation, ✗= Ablated) | | | | GPT-4o mini/Qwen-72B | | |
|---|---|---|---|---|---|---|
| **Prev. deals** | **Others. Prefer.** | **Candidates** | **Planning** | **5/6-way** | **6-way** | **Any** |
| ✓ | ✓ | ✓ | ✓ | 60 / 90 | 0 / 0 | 100 / 100 |
| ✓ | ✓ | ✓ | ✗ | 55 / 95 | 10 / 0 | 95 / 95 |
| ✓ | ✓ | ✗ | ✓ | 60 / 90 | 0 / 0 | 85 / 90 |
| ✓ | ✓ | ✗ | ✗ | 45 / 80 | 10 / 0 | 95 / 100 |
| ✓ | ✗ | ✓ | ✓ | 5 / 90 | 0 / 0 | 10 / 90 |
| ✓ | ✗ | ✓ | ✗ | 35 / 85 | 10 / 5 | 80 / 90 |
| ✓ | ✗ | ✗ | ✓ | 30 / 90 | 10 / 0 | 85 / 95 |
| ✓ | ✗ | ✗ | ✗ | 55 / 85 | 15 / 0 | 85 / 100 |
| ✗ | ✓ | ✓ | ✓ | 60 / 85 | 10 / 0 | 90 / 95 |
| ✗ | ✓ | ✓ | ✗ | 60 / 80 | 15 / 5 | 95 / 90 |
| ✗ | ✓ | ✗ | ✓ | 55 / 90 | 10 / 10 | 90 / 90 |
| ✗ | ✓ | ✗ | ✗ | 25 / 80 | 5 / 0 | 75 / 95 |
| ✗ | ✗ | ✓ | ✓ | 50 / 90 | 0 / 0 | 75 / 100 |
| ✗ | ✗ | ✓ | ✗ | 45 / 80 | 10 / 5 | 95 / 95 |
| ✗ | ✗ | ✗ | ✓ | 55 / 90 | 15 / 0 | 90 / 95 |
| ✗ | ✗ | ✗ | ✗ | 65 / 95 | 15 / 0 | 90 / 100 |

Table 2: Ablation study results. The ablation configuration used in original work for GPT-4 is highlighted in red. Large variance in performance is visible across configurations.

The results from **Experiment 2** can be found in Table 2, which demonstrates that the impact of ablation configuration varies depending on the model, highlighting the complexity of generalizing performance across different setups. Our goal with this extension is to investigate whether selecting a single ablation configuration based solely on its performance for one model might bias comparative evaluations. For instance, a configuration that leads one model to outperform another could yield opposite results when using a different configuration. Therefore, any rigorous comparison between models should explicitly account for the choice of ablation settings, as these significantly influence performance outcomes.

| | GPT4 (Abdelnabi et al., 2024) | | | GPT-4o mini | | | Mistral-Small | | | Qwen2.5-72B | | |
|---|---|---|---|---|---|---|---|---|---|---|---|---|
| | 5/6-way | 6-way | Any% | 5/6-way | 6-way | Any% | 5/6-way | 6-way | Any% | 5/6-way | 6-way | Any% |
| Base | 81 | 33 | 100 | 55 | 5 | 90 | 80 | 0 | 100 | 85 | 0 | 95 |
| Base_rewritten | 86 | 24 | 100 | 55 | 0 | 80 | 60 | 15 | 95 | 95 | 5 | 100 |
| Game 1 | 65 | 10 | 85 | 35 | 5 | 75 | 25 | 5 | 75 | 80 | 30 | 95 |
| Game 2 | 70 | 40 | 90 | 15 | 15 | 30 | 40 | 40 | 85 | 30 | 15 | 60 |
| Game 3 | 86 | 81 | 95 | 55 | 55 | 95 | 0 | 0 | 50 | 80 | 80 | 100 |

Table 3: Performance of models across different games.

The results of **Experiment 3** are in Table 3. Model performance varies inconsistently across games. For example, while Mistral-Small achieves competitive results in **games 1** and **2**, its comparatively low score on **game 3** suggests task-specific limitations. Similarly, GPT-4o mini and Qwen2.5-72B struggle most with **game 2**, whereas GPT-4 and Mistral-Small find **game 1** the hardest. These disparities highlight that difficulty is linked to unseen factors in game-agent dynamics.

From Table 3, it is clear that a game's relative difficulty can vary significantly depending on the model evaluated. In other words, a game that poses challenges for one model may be easier for another, and vice versa. This variability suggests that difficulty is not consistently aligned across models. Consequently, this raises important questions regarding the validity of using one negotiation game or another as benchmarks when evaluating language models' capabilities.

A subset of results for **Experiment 4** can be seen in Figure 1. To illustrate differences in score evolution, we display charts of a poor-performing and a high-performing model (in terms of game fulfillment).

In Figure 1(a) the USW for a poor-performing model is shown to decrease over time. There is a strong negative slope of -5.8 along with a high variance of 243. The combination of a steep negative gradient and

high variance suggests significant instability in the negotiation process. Additionally, the low correlation coefficient indicates that while a general downward trend exists, negotiation outcomes remain inconsistent. In the case of Phi-3.5-mini, the high variability and decreasing performance over time can be attributed to conversational drift, where negotiations break down as the scope of the conversation increases (Abdin et al., 2024a).

For high-performing models, such as that shown in Figure 1(b), there is a lower variance, 58.2, compared to the poor-performing model, along with a positive slope of 0.877. This, combined with a relatively high correlation coefficient of 0.687, suggests more stable and linear behavior in negotiation outcomes.

Finally, the results from the ESW and Nash Equilibrium metrics largely mimic the pattern of the USW metric, verifying that one model is not exclusively dominating the rounds.

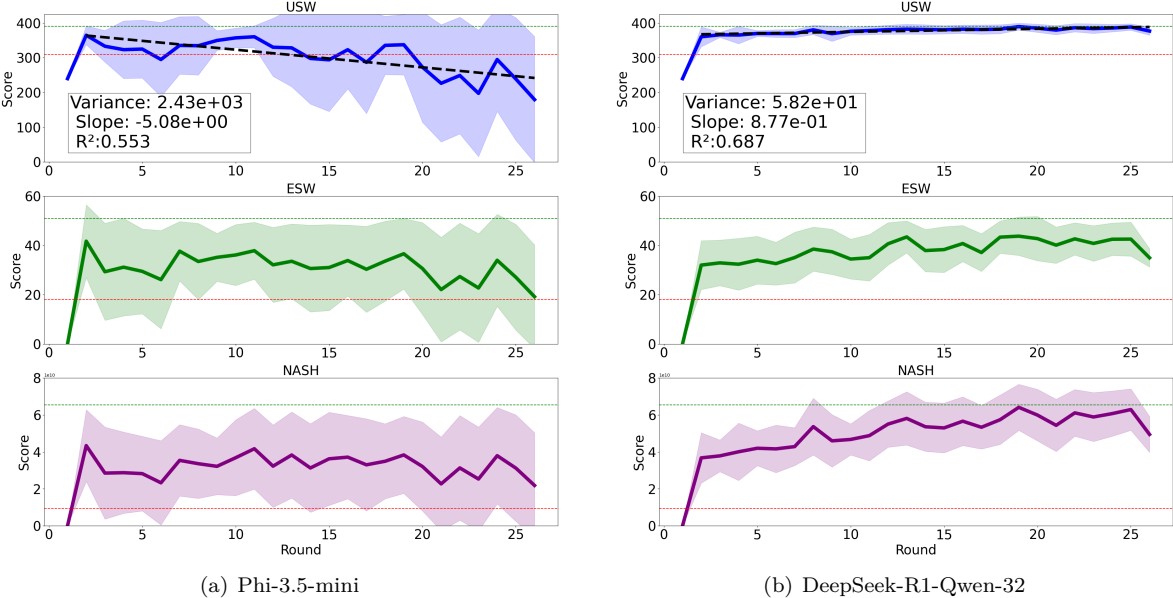

(a) Phi-3.5-mini
(b) DeepSeek-R1-Qwen-32

Figure 1: Evolution in different USW (blue), ESW (green), and NSW (red) metrics over time across two different models. The solid lines represent the mean score over time. The dashed colored lines represent the minimum and maximum achievable scores under a given metric. The black dashed line over USW is the linear least square regression.

The results for **Experiment 5** are displayed in Table 4. We compare the performance of Abdelnabi et al. (2024)'s baseline and our newly introduced baseline across all game variants. Our attempts to replicate their baseline did not yield results matching those reported in the original study. However, we provide an alternative, equally interpretable method demonstrating performance levels comparable to the original baseline.

|        | Original Baseline | | Our Baseline | |
|--------|---------|-------|---------|-------|
|        | 5/6-way | 6-way | 5/6-way | 6-way |
| Base   | 37      | 28    | 63      | 47    |
| Game 1 | 46      | 22    | 79      | 70    |
| Game 2 | 62      | 20    | 68      | 58    |
| Game 3 | 79      | 28    | 83      | 82    |

Table 4: Baseline proposed by Abdelnabi et al. (2024) and our baseline evaluated on different game variants.

Although our primary goal was to replicate the original baseline, our difficulty in doing so and the lack of a clear definition of importance prompted us to propose a new, fully reproducible baseline. This does not negate the value of the original method; rather, it underscores that multiple algorithmic baselines are feasible. Based on the results, which are comparable to those reported for the original paper, and the fact that our method offers both accessibility and reproducibility, our proposed method can serve as an effective, interpretable alternative.

In the context of this benchmark, which seeks to assess negotiation abilities, defining an appropriate baseline is nontrivial. Since a rule-based algorithm capable of solving the game can be readily implemented, the key challenge is not merely evaluating a model's mathematical ability to negotiate but capturing the subtleties of human negotiation, such as deception, bargaining, and strategic withholding of information. A rule-based approach serves as a plausible example of rational behavior in negotiation but may fail to reflect the nuanced decision-making observed in human interactions.

## 4.2   Adjustments claim

Regarding prompting the model to change settings, we find that the construction project setting appears not only in the five original games but also in all ten additional games generated using the prompt from the original work. This consistency suggests an inherent bias in the prompt, which always steers the negotiation scenario toward a construction project. By removing specific terms such as "project" and "resources", we observe a broader range of negotiation contexts emerging, revealing the extent of this bias. With our revised prompt (see Appendix G), we identify a more diverse set of negotiation settings, including e.g. discussions on military spending budgets for the upcoming year and planning a conference, where aspects like floor allocation and catering must be negotiated. The results clearly demonstrate that the negotiation scenarios generated with the original prompt are not diverse, and that adjustments are needed to obtain true diversity.

As for score function updates, Table 5 presents the results of **Experiment 6**, which examines the diversity of the provided games. From a structural standpoint, these metrics reveal that, although each configuration exhibits slight numerical differences, they fall within a fairly narrow range overall. We note that the fraction of acceptable deals $|\Pi_{\mathrm{acc}}|$ is almost identical for all games by design (Abdelnabi et al., 2024), which does not contribute to the claim that the games are diverse. Similarly, the fraction of hard-acceptable deals $|\Pi_{\mathrm{hard}}|$ remains relatively small across all cases. Sparsity shows moderate fluctuation, while the intersection-over-union measure ranges from about 18.75% to 29.77%, suggesting that, while differences exist in preference overlap and zero-valued options, they are not substantial. These results indicate that each setup retains comparable intrinsic characteristics.

|  | $|\Pi_{\mathrm{acc}}|/|\Pi|$ | $|\Pi_{\mathrm{hard}}|/|\Pi|$ | % Spar. | % IoU |
|---|---|---|---|---|
| Base | 55/720 | 12/720 | 38.6 | 18.75 |
| Base_rewritten | 55/720 | 12/720 | 38.6 | 18.75 |
| Game 1 | 57/720 | 21/720 | 23.68 | 29.77 |
| Game 2 | 57/720 | 18/720 | 29.82 | 25.75 |
| Game 3 | 55/720 | 35/720 | 42.98 | 21.65 |

Table 5: Score function statistics.

The fact that the computed statistics do not span the entire range of possible values (e.g., 0 to 100 for percentage-based metrics) is not, by itself, sufficient to conclude that they lack diversity. To establish that the games are diverse in terms of these statistics, a larger set of games should be generated, including cases where some statistics cover a wider section of possible values. These new games should be tested to determine whether they reach a saturation point in performance, either becoming trivially easy or outright impossible. If this occurs, it would indicate that including games with such statistics is redundant and that the full spectrum of meaningful values has been effectively covered.

The results for **Experiment 7** are shown in Figure 2, where we examine how easily adjustable games are using the modifiers provided by original authors. The results are consistent with the findings of the original paper: in Figure 2(a) performance tends to increase consistently with a decrease in the minimum threshold per player. Similarly, we find that more agreements are reached with a lower number of players, see Figure 2(b).

## 4.3   Behavioral claim

Table 6 presents the results of **Experiment 8**, reproducing the adversarial behavioral experiments. Consistent with Abdelnabi et al. (2024), we observe that performance declines when all players are greedy, presumably because mutual self-interest stalls compromise. Similarly, when $p_1$ is greedy, we also see dimin-

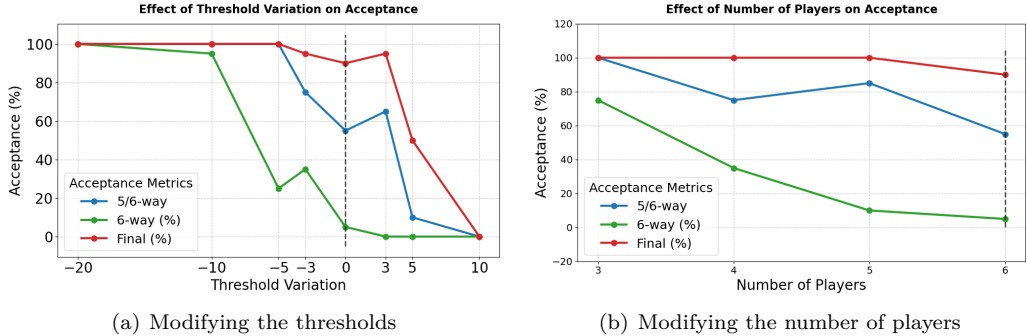

(a) Modifying the thresholds  (b) Modifying the number of players

Figure 2: Varying thresholds and number of players in **base game** for GPT-4o mini

ished performance, because $p_1$ wields veto power and makes the final proposal—thus it can block suboptimal agreements until it secures a favorable share.

We note a surprising result, where we find that when a particular player—namely the Environmental League—is greedy, GPT-4o mini and Qwen2.5-72B show an increase in 5/6-way deals. One possible explanation is that these models, when facing a single (but not first-mover) greedy party, adjust their negotiation strategy in ways that induce earlier concessions or more expansive coalitions among the other players—ultimately raising the rate of larger multi-party agreements despite the Environmental League's hardened stance.

| Variant | GPT-4o mini | | Qwen2.5-72B | | GPT4 * | |
|---|---|---|---|---|---|---|
| | 5/6-way | 6-way | 5/6-way | 6-way | 5/6-way | 6-way |
| All compromising | 55 | 5 | 85 | 0 | 81 | 33 |
| One greedy ($p_i \in P_{\text{const}}$) | 80 | 10 | 95 | 0 | 57 | 30 |
| One greedy ($p_1$) | 45 | 5 | 50 | 5 | 27 | 9 |
| Two greedy ($P_{\text{benefit}}$) | 95 | 5 | 75 | 0 | 65 | 15 |
| All greedy | 50 | 0 | 15 | 5 | 26 | 11 |
| Adversarial (untargeted) | 90 | 0 | 85 | 0 | 63 | – |
| Adversarial (targeted) | 70 | 5 | 85 | 0 | 58 | – |

Table 6: Performance in the different behavioral variants, comparing GPT-4o mini, Qwen2.5-72B, and GPT4. $P_{\text{benefit}}$ is the set of players whose goals align to those of $p_1$ and $P_{\text{const}}$ is similarly defined but with neutral goals with respect to $p_1$. It is not clear how these sets are delineated in the original paper, so we replicated the setup with the players indicated in the open source configurations. The results for GPT4, marked with *, were taken from the original work.

We also reproduce trends in game evolution with respect to the players of interest (i.e., the greedy, adversarial, and targeted players) in Figure 3. For the cooperative case, we observe that *Environmental League* is more willing to compromise, as indicated by its lower utility. Conversely, when prompted to adopt a greedy strategy, the affected agent maintains a higher individual score. In the adversarial experiments, the utility for *Local Labour Union* is lower in the targeted case, demonstrating the impact of adversarial play.

These trends become apparent when comparing model performance. In the $p_i$ greedy setting, the last round's distance between the greedy agent and the aggregated USW was 12.86 for GPT-4o mini but significantly higher (24.85) for Qwen2.5-72B-Instruct, indicating that Qwen enforces greedy behavior more strongly. Similarly, in the untargeted adversarial setting, the distance between the would-be targeted agent and aggregated USW was -33.91 for GPT-4o mini, whereas for Qwen2.5-72B-Instruct, it was only -14.03. These results indicate that stronger models like Qwen2.5-72B-Instruct better capture prompt-induced behavior.

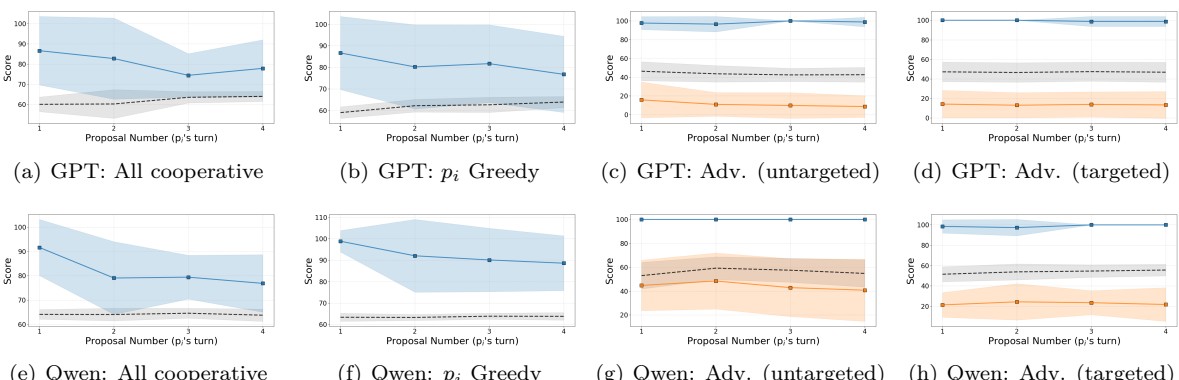

(a) GPT: All cooperative  (b) GPT: $p_i$ Greedy  (c) GPT: Adv. (untargeted)  (d) GPT: Adv. (targeted)

(e) Qwen: All cooperative  (f) Qwen: $p_i$ Greedy  (g) Qwen: Adv. (untargeted)  (h) Qwen: Adv. (targeted)

Figure 3: Evolution of deals for several behavioral configurations. *Environmental League* is $p_i$ and acts as greedy for the greedy experiment and as adversarial in both the untargeted and targeted experiments. For the latter, *Local Labour Union* is the target.

## 5  Discussion

The benchmarking framework proposed by Abdelnabi et al. (2024) comprises a complex simulation environment that evaluates LLM negotiation. By choosing a more complicated environment model based on a negotiation game used to teach humans, they overcome a problem inherent in related works (Akata et al., 2023; Brookins and DeBacker, 2023): that the simulated games are not complex enough to represent reality. Furthermore, utilizing LLMs in a negotiation setting designed for humans opens the door to analyzing human-LLM interaction through the lens of behavioral game theory. Additionally, the extensibility of the framework helps to maintain its relevance in this era of increasingly powerful LLMs. However, this approach also comes at a cost.

One key challenge is that, due to its complexity and extensibility, Abdelnabi et al. (2024) introduces a potentially infinite number of comparison axes through their various games, which makes comparative performance across multiple games difficult to interpret, as demonstrated in our experiments, particularly in **Experiment 3**. Additionally, it complicates the application of established formalisms, such as Nash Equilibrium analysis, which is commonly used in other benchmark studies (Brookins and DeBacker, 2023). The absence of such formal criteria further hinders the objective assessment of benchmark results. In our evaluation of the **Benchmark Claim** and **Adjustments Claim**, we have demonstrated various practical examples of this difficulty in interpreting benchmark results. This includes factors such as our ablation analysis in **Experiment 2**, and the demonstrated weaknesses in diversity of the provided games in **Experiment 6**. Through our extensions, we have provided additional context to the benchmark such as transparency on the previously mentioned factors as well as metrics from social welfare to assist a user in further understanding the benchmark results. Additionally, while the various minor bugs in the original implementations do not invalidate the results of the original paper, they potentially pose problems to users evaluating weaker models. By fixing them, we increase the potential audienceof this benchmark.

One potential idea for future work is developing metrics capable of describing and categorizing the various components of different negotiation games. A particularly useful example within the current framework would be a categorization of games that aligns with model performance. This would allow model developers to claim, for instance, that their new LLM outperforms others in specific game categories but not in others. Another exciting avenue of future work is to explore benchmarking the adversarial behavior of the agents through the framework by creating a "robustness against adversarial attack" metric, in order to quantitatively compare the resilience in the adversarial setting introduced in Abdelnabi et al. (2024). An example of a potential metric could be the percentage of 5-way successful games, weighted by sparsity, in an adversarial environment with a single agent causing issues. We leave these potential works as ideas for further increasing the utility of this benchmark.

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

## Acknowledgments

This publication stems from research conducted as part of the Master's Programme in Artificial Intelligence at the University of Amsterdam. We give our thanks to the university, and in particular the teaching and organizing staff in the FACT course where this paper originated. Ioannis Kapetangeorgis was supported by the John S. Latsis Public Benefit Foundation postgraduate scholarship. Special thanks goes to Alexandre da Silva Pires, who supervised our first paper-writing experience and without whom this work would not have been possible.

## Appendix

## A    Implementation details

### A.1    Model Details

We describe the models used in our experiments in Table 7. Some models were distilled from a larger model, as shown in column 3, and some models were also quantized for quicker run time considering computation constraints. The level of quantization was primarily determined by computational and memory constraints; whenever feasible, a less aggressive quantization (i.e. int8) was employed, though larger models necessitated int4 quantization. Details can be found in Table 8.

| Model | Parameters (B) | Architecture | Context Window | Additional Details |
|---|---|---|---|---|
| Llama-3.3-70B | 70 | Decoder-only Transformer | 128k tokens | SwiGLU, RoPE, RMSNorm |
| Mistral-Small | 22 | Cost-effective Transformer | 33k tokens | Task-optimized |
| Phi-3.5-mini | 3.8 | Compact Transformer | 4k tokens | Efficiency-focused |
| Mistral-8B | 8 | Performance-optimized | 128k tokens | Balanced efficiency |
| Qwen2.5-72B | 72 | Dense decoder-only | 131k tokens | Advanced capabilities |
| Phi-4 | 14 | State-of-the-art Transformer | 16k tokens | High performance |
| Qwen2.5-7B | 7 | Efficient Transformer | 131k tokens | Diverse applications |
| Llama-2-13B | 13 | Decoder-only Transformer | 4k tokens | SwiGLU, RoPE, RMSNorm |
| Mixtral-8x7B | 56 (8x7) | Mixture of Experts | 33k tokens | Computational efficiency |
| Llama-3-8 | 8 | Decoder-only Transformer | 8k tokens | SwiGLU, RoPE, RMSNorm |
| DeepSeek-R1-Distill-Qwen-32B | 32 | Distilled from DeepSeek-R1, based on Qwen2.5-32B | 32,768 tokens | Fine-tuned with DeepSeek-R1-generated data for reasoning |
| DeepSeek-R1-Distill-Llama-70B | 70 | Distilled from DeepSeek-R1, based on Llama-3.3-70B-Instruct | 128,000 tokens | Fine-tuned with DeepSeek-R1-generated data for reasoning |

Table 7: Summary of open-source models evaluated, including their parameters, architecture, context window size, and additional details.

.

| Model | Quantization Applied |
|---|---|
| Llama-2-13B | int8 |
| Llama-3.3-70B | int4 |
| Qwen2.5-72B | int4 |
| Phi-4 | int8 |
| Mistral-Small | int8 |
| Mixtral-8x7B | int4 |
| DeepSeek-R1-Distill-Qwen-32B | int8 |
| DeepSeek-R1-Distill-Llama-70B | int4 |

Table 8: Quantized models and techniques applied

.

### A.1.1    Code Structure

Both the original and our codebase are structured into multiple Python scripts, each handling a specific aspect of the experimental process. The main script, main.py, serves as the entry point, integrating all components

to execute experiments. Supporting modules manage agent behavior, negotiation rounds, prompt generation, and data handling. Detailed documentation of the code structure is available in the README file of the GitHub repository.

Initially, configuration options were managed through a combination of command-line arguments and external configuration files. The command-line arguments allow users to specify parameters such as temperature, number of agents, number of issues, number of rounds, game name, and model name. Additionally, each game directory contains a *config.txt* file that defines the setup for each agent, including their name, associated files, role, incentive, and model. This structure enables customization of experiments by modifying both command-line inputs and configuration files.

However, to further enhance configurability and streamline the experimental workflow, several improvements have been introduced beyond this initial approach. Previously, the specification of models and incentives for agents was handled by modifying the game's configuration file. However, due to the frequent need to rerun the same game with different model and incentive configurations, the *–model* and *–incentive* command-line parameters have been added to simplify the process and reduce manual intervention. Additionally, the *–quantization* parameter has been introduced to enable quantization for Hugging Face models, improving performance and reducing memory requirements. To address concerns regarding sensitive information disclosure during negotiations, the *–restrict_leakage* flag has been implemented, with further details provided in the respective section of the report. A *–dry_run* option has also been included to facilitate testing and debugging by disabling actual API calls to language models, allowing users to inspect prompts without incurring computational costs. Furthermore, environmental sustainability considerations have been incorporated into the framework through the integration of the *codecarbon* EmissionsTracker, which records the carbon footprint of experiments, with the option to specify project names via the *–emission_project* parameter. These enhancements build upon the initial configuration system, providing a more efficient, scalable, and environmentally conscious framework for conducting negotiation experiments.

## A.2   Computational Details

**Hardware Setup**   The experiments were conducted using local machines equipped with NVIDIA A100 and A6000 GPUs, with memory capacities of 40 GB and 48 GB, respectively. The computational resources were sufficient to accommodate all models tested, with the most demanding model requiring a maximum of 48 GB of VRAM. Thus, the experiments can be reproduced on a system with similar GPU specifications.

**Runtime Measurements**   The runtime for each negotiation session varied depending on the model used. Table 9 presents the average runtime per model for a complete negotiation session comprising 24 rounds. It was observed that computational resource demands varied significantly among different models, with some lightweight models requiring considerably fewer resources compared to larger models.

| Model | Runtime (min) |
|---|---|
| GPT-4o mini | 4 |
| Mistral-Small-Instruct-2409 | 20 |
| Phi-3.5-mini-instruct | 6 |
| Llama-3.3-70B-Instruct | 22 |
| Ministral-8B-Instruct-2410 | 15 |
| Phi-4 | 15 |
| Qwen2.5-7B-Instruct | 9 |
| Llama-2-13b-chat-h | 11 |
| Mixtral-8x7B-Instruct-v0.1 | 8 |
| Meta-Llama-3-8B-Instruct | 3 |
| Qwen2.5-72B-Instruct | 29 |
| DeepSeek-R1-Distill-Qwen-32B | 68 |
| DeepSeek-R1-Distill-Llama-70B | 65 |

Table 9: Average runtime per negotiation session (24 rounds) across different models.

The total time required to complete all experimental runs, including different game variants, amounted to 422 hours. For open-source models, the total GPU/CPU hours required were 386 hours. For GPT-4o mini, the OpenAI API was utilized, resulting in different resource consumption patterns.

**Storage Requirements**   The primary storage requirement for the experiments was to store the downloaded pretrained model weights. By storing one model at a time, the total storage demand was limited to approximately 300 GB. All experiment outputs and logs were stored for post-hoc evaluation and further analysis.

**Software Environment**   The experiments were conducted on Linux-based systems. For OpenAI models (i.e., GPT-4o and GPT-4o mini), the OpenAI API was utilized, whereas for open-source models, the Hugging Face `transformers` library was employed. Additionally, the `codecarbon` library was used to monitor carbon emissions during the computational processes.

## B   Ablation prompts

We present here the prompts used to create the Chain of Thought for the agents. These variations were used to recreate and extend the ablation study, see Table 2.

### B.1   Previous Deals

The prompt to consider previous deals is:

```
 Please use the scratchpad for the following:  Given the history of previous suggested
deals, calculate your scores for each past suggested deal in the history.  Then, use
these deals to further reason about your next proposed deal.
```

### B.2   Others Preferences

The prompt to consider the preferences of other players is:

```
In your scratchpad, 1) think about what others may prefer, 2) Based on others'
preferences and history and your notes, propose one proposal that balances between your
scores and accommodating others and that is more likely to lead to an agreement.
```

### B.3   Candidates

The prompt to generate three deal candidates before choosing one is:

```
Please use the scratchpad for the following:  Before suggesting a final deal, display
three possible deals to suggest while keeping in mind your objectives, and number them
with (1), (2), and (3).  When your final deal is made, choose one of these three deals.
```

### B.4   Planning

The prompt to release a plan at each round is:

```
After the final answer, building on your current move and analysis, briefly write down
short notes for yourself of what exact options you can explore the next time you speak.
Enclose the notes between <PLAN> and </PLAN>.
```

## C   Carbon Emissions

Environmental sustainability considerations have been incorporated into our framework through the integration of the `codecarbon`[¶] emissions tracker, which records the carbon footprint of experiments by considering the specifications of the system running the model, the carbon efficiency of the country it is powered in, and electricity usage. By extrapolating the results of running the carbon tracker for each model on a set of 20 games, the total cost of our runs are approximately $48.7 kg\ CO_2e$. We used 24 million input and 28 million output tokens of GPT-4o mini. We now state the carbon emissions that our experiments entailed.

We use the figures from Table 10 to extrapolate the total carbon usage per experiment based on which models are run.

$$\text{Total Carbon Emissions} = \text{Carbon}(T_1) + \text{Carbon}(T_2) + \text{Carbon}(ablation) + \text{Carbon}(T_6) + \text{Carbon}(T_{10}) \quad (1)$$

---

[¶]https://codecarbon.io/

| Model | Average Emissions (kg $CO_2$e) |
|---|---|
| Llama-2-13B | 0.4 |
| Llama-3.3-70B | 1.0 |
| Meta-Llama-3-8B | 0.1 |
| Ministral-8B | 0.6 |
| Mistral-Small | 0.7 |
| Mixtral_8x7B | 0.3 |
| Phi-3.5-mini | 0.2 |
| Qwen2.5-72B | 1.3 |
| Qwen2.5-7B | 0.4 |
| Phi-4 | 0.6 |
| Deepseek-R1-Distill-Llama-70B | 2.7 |
| Deepseek-R1-Distill-Qwen-32B | 2.8 |

Table 10: Estimated average carbon emissions ($kg\ CO_2e$) per model for processing a batch of 20 games
.

where $T_n$ corresponds to the carbon emissions associated with the experiments run to populate Table $n$. Since all runs are done with 20 games, these calculations are straightforward and the code is provided. Additionally, we monitored the emissions for running the ablation study and found that

$$\text{Carbon(ablation)} = 19.8 kg\ CO_2e$$

The carbon generated per table is

$$\text{Carbon}(T_1) = 11.1$$
$$\text{Carbon}(T_2) = 2.0$$
$$\text{Carbon}(T_6) = 8.0$$
$$\text{Carbon}(T_{10}) = 7.8$$

And thus the final calculated carbon output is

$$\text{Total Carbon Emissions} = 48.7 kg\ CO_2e \tag{2}$$

These emissions are approximately equivalent to those produced by driving a Ford F-150 for 117 miles, which is roughly the distance between Rotterdam and Groningen.

We also used GPT-4o mini. However as the running cost for running this model is unknown, we exclude it from the calculation.

# D   Leakage Results

| Model | Original Code | | | | | Our Code | | | | | |
|---|---|---|---|---|---|---|---|---|---|---|---|
| | **Final ↑ (%)** | | **Any ↑ (%)** | **Wrong ↓ (%)** | **Leaked ↓ (%)** | **Final ↑ (%)** | | **Any ↑ (%)** | **Wrong ↓ (%)** | **Leaked ↓ (%)** | **Failed ↓ (%)** |
| | **5-way** | **6-way** | | | | **5-way** | **6-way** | | | | |
| Llama-2-13b | 30.0 | 0.0 | 75.0 | 17.92 | 9.23 | 8.33 | 0.0 | 25.0 | 25.51 | 0.32 | 40 |
| Llama-3-8B | 25.0 | 0.0 | 70.0 | 8.14 | 69.42 | 35.0 | 0.0 | 55.0 | 9.8 | 0 | 0 |
| Ministral-8B | 25.0 | 0.0 | 50.0 | 12.88 | 7.12 | 22.22 | 0.0 | 50.0 | 11.11 | 0 | 10 |
| Mixtral-8x7B | 30.0 | 5.0 | 55.0 | 23.64 | 24.81 | 26.67 | 0.0 | 66.67 | 20.14 | 0 | 25 |
| Phi-3.5-mini | 10.0 | 10.0 | 50.0 | 12.87 | 40.77 | 35.0 | 10.0 | 95.0 | 10.31 | 0.38 | 0 |
| Qwen2.5-7B | 65.0 | 25.0 | 100.0 | 11.15 | 44.62 | 44.44 | 5.56 | 88.89 | 8.76 | 0 | 10 |

Table 11: Comparison of model performance metrics before and after the leakage issue is resolved.

| | Original Code | | | Our code | | |
|---|---|---|---|---|---|---|
| **Model** | **Successful Games** | **Leaked Games** | **Leaked (%)** | **Successful Games** | **Leaked Games** | **Leaked (%)** |
| Llama-2-13b | 20 | 11 | 9.23 | 12 | 1 | 0.32 |
| Llama-3-8B | 20 | 20 | 69.42 | 20 | 0 | 0.00 |
| Ministral-8B | 20 | 20 | 7.12 | 18 | 0 | 0.00 |
| Mixtral-8x7B | 20 | 20 | 24.81 | 15 | 0 | 0.00 |
| Phi-3.5-mini | 20 | 20 | 40.77 | 20 | 2 | 0.38 |
| Qwen2.5-7B | 20 | 20 | 44.62 | 18 | 0 | 0.00 |

Table 12: A comparison of models that are prone to leakage before and after code adjustments.

After applying our method to the original experiments and manually verifying outcomes, we find that our strategy significantly reduces leakage, lowering the associated metrics by up to 69.42% compared to the original approach. Furthermore, our refined method results in lower 5/6-way agreement rates, as agents no longer have unintended access to private information that could facilitate deal-making. While no detection framework is entirely exhaustive, our improvements offer a substantial reduction in risk of leakage.

# E   Social Welfare Definitions

Let $S_m$ denote a score function such that each deal is assigned a natural number value under some utility $m$; $S_m(\pi) : \Pi \to \mathbb{N}$.

1. **Utilitarian Social Welfare (USW):** Aggregates each player's utility for a deal, as in the original study, but does not consider fairness.

$$S_{\text{usw}}(\pi) = \sum_{p_i \in P} U_{p_i}(\pi) \tag{3}$$

2. **Egalitarian Social Welfare (ESW):** Ensures all players benefit by taking the minimum utility across players for each deal.

$$S_{\text{esw}}(\pi) = \min_{p_i \in P} \big\{ U_{p_i}(\pi) \big\} \tag{4}$$

3. **Nash Social Welfare (NSW):** Promotes fairness by multiplying players' utilities, favoring even distributions (e.g., $4 \times 4 > 2 \times 6$).

$$S_{\text{nsw}}(\pi) = \prod_{p_i \in P} U_{p_i}(\pi) \tag{5}$$

## F    IoU Equations

The Intersection over Union (IoU) metric quantifies similarity between agent scoring functions by measuring the overlap in their preferences. Pairwise IoU computes the ratio of shared valuations between two agents, while the overall IoU aggregates these values across all agent pairs. The following equations formally define these computations, as referenced in **Experiment 6**, to assess preference alignment and diversity.

$$\text{IoU} = \frac{1}{|P|} \sum_{p_x \in P} \frac{1}{|P \setminus \{p_x\}|} \sum_{p_y \in P \setminus \{p_x\}} \text{Pairwise IoU}(p_x, p_y) \tag{6}$$

$$\text{Pairwise IoU}(p_x, p_y) = \frac{1}{|I|} \sum_{i=1}^{|I|} \frac{\sum_{j=1}^{|I_j|} \min(p_{x,i_j}, p_{y,i_j})}{\sum_{k=1}^{|I_k|} \max(p_{x,i_k}, p_{y,i_k})} \tag{7}$$

## G    Alternative Prompt

```
 You are an expert in negotiation games and have read many books on the subject.  Please
help me in creating a negotiation game.  The game consists of 6 players (party 1,
party 2, party 3, etc.)  who are negotiating over 5 issues.  Each of the 5 issues has
different sub-options (2 issues have 3 options, 2 issues have 4 options, 1 issue has 5
options).  One of the players makes a first proposal.  The issues involve the impact of
the negotiated outcome on stakeholders.  The other players represent different parties
with competing interests.  The parties must not include a mediator.  The issues represent
the interests of other parties.  The issues do not necessarily have a one-to-one mapping
to each party; different parties might have similar or competing interests under each
issue (e.g., one wants more funding, one wants less funding, etc.).  Some parties do
not care at all about certain issues (they only care about a subset of issues).  The
game is based on cooperative bargaining.  Your task is to create the background story
and the role of each party according to the previously mentioned guidelines.  Please
indicate their general goals and motivations and their objectives from the negotiation.
You should also create the issues they are negotiating over (please name them issues A,
B, etc.)  by specifying the different sub-options (A1, B1, C1, etc.).  For each issue,
please specify what the preferences of each of the parties are over the issues and why
they prefer so (e.g., Party 1 prefers A3 then A2 then A4, etc.).  Please also assign
priorities of the issues to each party and explain why (e.g., Party 1 cares the most
about issue A, they do not care about issue D). Please also indicate if an issue is
much more important than the others.  Make it interesting with lots of potential for
cooperation and competition between parties!!  Make the issues and options have some
implications over generally more than one party involved, but you can have some parties
with no interest at all in some issues.  Remember that it is a cooperative non-zero-sum
game.
```

