# OpenReview forum: "[Re] Benchmarking LLM Capabilities in Negotiation through Scoreable Games"
_TMLR — Accepted by TMLR_

### Review · Reviewer_KzXy · 2025-03-22

**Summary Of Contributions:**

Find some technical flaws of a paper published in NeurIPS 2024 D&B Track. In conclusion:

1. We need to consider prompt sensitivity in such experiments.
2. We need to have an objectively defined difficulty of games instead of using the performance on one model to define.
3. We need to handle LLM-generated test cases with care, especially on the bias issue.

**Audience:**

Yes

**Claims And Evidence:**

Yes

**Requested Changes:**

1. Add more details to section 3.1.5. This section plays an important role to show the flaws in the original paper. However, it is hard for readers who are not knowledgable about the original paper to understand the logic here. For example, what are the <ANSWER>, <plan>, <scratchpad> tags? What are they used for? "The system inadvertently exposes confidential outputs": Why and how does the system expose such information?
2. Different experiments use different models currently. Table 2 uses GPT-4o-mini and Qwen-72B, Table 3 uses GPT-4, GPT-4o-mini, Mistral-Small, Qwen-2.5-72B. Figure 1 uses Phi and R1. I think it is better to have a unified setting or explain clearly why certain models are selected for different tasks.
3. Discuss whether quantization will affect the model performance. Explain why some models use int4 while others use int8.
4. Add some results on stronger closed-source models such as GPT-4o, Gemini-2.0, or Claude-3.7.
5. Add deeper analysis about how the fixed bugs may have influence on the original results & conclusions.

**Strengths And Weaknesses:**

Strengths
- Extensive experiments: These experiments include replication of prior results, extended ablation studies, behavioral tests, and the introduction of new evaluation frameworks.
- Technical improvements: It fixes key flaws in the original benchmark, such as leakage bugs and inconsistent scoring logic, improving fairness and reproducibility.
- New evaluation metrics: Introduces social welfare-based metrics (Utilitarian, Egalitarian, Nash) to better capture negotiation quality and fairness.

Weaknesses
- Although the study highlights that different games present varying challenges to different models, it fails to offer a clear, standardized metric for defining or comparing game difficulty.
- The rationale for selecting specific models is not always clear or consistently justified (see Requested Changes below).

---

> ### Author Response · Authors · 2025-04-08
>
> We appreciate the thoughtful feedback and have revised the manuscript accordingly. For clarity, we address each point separately:
>
> **1. On the Lack of a Standardized Metric for Game Difficulty**:
>
>  Although the study highlights that different games present varying challenges to different models, it fails to offer a clear, standardized metric for defining or comparing game difficulty.
>  We appreciate the reviewer’s concern regarding the absence of a standardized metric. While we agree that a standardized metric could provide additional clarity, our primary goal was to demonstrate that the original benchmark itself is ambiguous. By showing that model performance varies significantly across games, we argue that this variance—without the need for an external, unified metric—already highlights inconsistencies in the benchmarking process. Our analysis intends to underline the point that even without a standardized difficulty metric, the original benchmark’s ambiguity becomes evident.
>
> **2. Clarification and Expansion of Section 3.1.5**:
>
>  Section 3.1.5 is critical for illustrating the flaws in the original paper but is difficult for readers unfamiliar with that work to follow. For example, the roles of <ANSWER>, <plan>, and <scratchpad> tags and how the system exposes confidential outputs are not clearly explained.
>
>  We have rewritten Section 3.1.5 (Code Adjustments) to include a more detailed description of the key components:
> We now explicitly explain that <ANSWER>, <plan>, and <scratchpad> tags are used to delineate different parts of the model’s response generation process. Specifically, these tags help separate the final answer from intermediate reasoning steps.
>
>
> Additionally, we clarify that the system exposes confidential outputs due to the way these tags are processed and logged in the original implementation. The revised text now outlines both the “how” and the “why” behind this exposure, making the logic accessible to readers who may not be deeply familiar with the original paper.
>
>
>
> **3. Justification for Model Selection Across Experiments**:
>
>  Different experiments currently use different models (e.g., Table 2 uses GPT-4o-mini and Qwen-72B, Table 3 uses GPT-4, GPT-4o-mini, Mistral-Small, Qwen-2.5-72B, and Figure 1 uses Phi and R1). A unified setting or clear explanation for these choices is necessary.
>  We acknowledge the need for clarity regarding model selection. In our revision, we have expanded the manuscript to justify our decisions:
> The diversity of models was chosen intentionally to test our approach across a wide range of settings.
>
>
> We now provide a clearer explanation for why certain models were selected for specific tasks, emphasizing that the differences in performance across these selections underscore the inherent challenges of the original benchmark.
>
> This explanation helps to contextualize the experimental results and supports our argument about the ambiguity of the benchmark.
>
>
>
> **4. Discussion on Quantization Effects**:
>
>  The manuscript should discuss whether quantization affects model performance and explain why some models use int4 while others use int8.
>
>  In response, we have added a concise discussion on quantization within the revised manuscript. Our changes include:
> Adding in the appendix an explanation to the rationale behind using different quantization levels (int4 vs. int8) based on computational constraints.
>
>
> Briefly outlining how quantization may impact performance and what implications this has for benchmarking, thereby offering additional context for the reader.
>
>
>
> **5. Inclusion of Results from Stronger Closed-Source Models**:
>
>  It is suggested to add some results on stronger closed-source models such as GPT-4o, Gemini-2.0, or Claude-3.7.
>
>  While we agree that incorporating results from closed-source models would further strengthen the study, we regret that resource constraints prevent us exploring that. We have however added to Table 1 results for GPT-4o.
>
> ----------------------------------------------------
>
> We trust that these revisions address the reviewer’s concerns and clarify the contributions and methodology of our study. Thank you for your constructive feedback.

---

### Review · Reviewer_aE6q · 2025-03-26

**Summary Of Contributions:**

The paper reproduces the results of the paper by Abdelnabi et al. named “Cooperation, competition, and maliciousness: LLM-stakeholders interactive negotiation” published in NeurIPS 2024 Dataset and Benchmark track. Additionally, the paper updates the originally used models in the benchmark with recently released ones and fixes minor issues in the original code. The paper also expands the discussion on the benchmark itself and provides some additional context on the negotiation aspect of it. Finally, the paper highlights some future directions to expand the metrics used in the original benchmark and the manuscript itself.

**Audience:**

No

**Broader Impact Concerns:**

There are no broader impact concerns in my opinion.

**Claims And Evidence:**

No

**Requested Changes:**

There are several changes that could enhance the paper in matching the claims proposed in the abstract and introduction:

•	The abstract, in itself, is ambiguous and could be made more concrete. For example, explicitly mentioning how is model comparison is ambiguous in the original benchmark and how did this manuscript make it clearer or objective. Additionally, some of the terms such as leakage detection and ablation transferability are also vague and do not clearly address the issues or highlighting how does this paper improves that. In general, the abstract is neither clear about the exact parts that need rectification in the original benchmark, nor the exact contributions made by this work other than extending the models and discussion (which is the legitimate use case for a benchmark paper and not a judge of reproducibility).

•	The discussion does not match what is mentioned in the abstract and introduction. While the abstract mention critical issues with the original manuscript, the discussion mentions minor changes and minor fixes. This is also apparent across most of the experiments presented in the manuscript itself. The manuscript also seems to mix between design issues and coding/mistakes in the original benchmark which should be rectified (see weaknesses section for more details).

•	The contributions section should be rectified as well to match the actual contribution and what is presented in the experiments and discussion (see weaknesses section as well).

•	Additional metrics should be provided as mentioned in the abstract to justify the need for a reproducibility check in the first place as the manuscript is currently showing that the original benchmark already provides sufficient metrics for evaluation as seen in Table 1. In other words, additional metrics (as suggested for future work in the discussion section) should be the focus of this manuscript as a reproducibility paper and as suggested in the abstract. Otherwise, the additional metrics added such as the sparsity and IoU metrics seems to measure the original benchmark itself and not a metric that could be used to enhance the benchmark when judging the agents.

**Strengths And Weaknesses:**

Strengths:

•	The paper extends the original benchmark with recent models.

•	The paper fixes some minor issues in the original code such as using greater than and equal consistently across experiments.

Weaknesses:

•	The major weakness is that the manuscript confuses design issues and plausible extensions with bugs in implementation. This is apparent across the entire manuscript as follows:

o	The Addressing Leakage bug: doing a postprocessing approach to filter out the leakage or enabling smaller models to perform better is an extra engineering process that is now a confounding factor that should be benchmarked on its own. This is the opposite to actually benchmarking the models on their own accord in handling such nuances without heavy post processing. This also completely contradicts the claim made in the manuscript “An analysis of the original implementation reveals that the reported leakage metrics are largely due to code limitations rather than intrinsic model behavior”. What is explained after in the same section shows that limitation is actually an intrinsic model behavior and the paper method to mitigate it is an engineering approach and design choice to overcome such a limitation. This is opposite to other benchmarking methods such as the original benchmark and [1] that consider the hallucination and illogical behavior as part of the benchmarking aspect

o	The adjustment claim: While this manuscript shows a possible context bias in generating new games, it did not contradict the original negotiation claim about assessing the different models. For example, in table 3, in can be seen that Qwen2.5-72B is already performing better than GPT4o-mini and Mistral-small.

[1] Bianchi, Federico, et al. "How well can LLMs negotiate? NEGOTIATIONARENA platform and analysis." @ICML

•	While the manuscript provided numerous future work prospects in the discussion section. These is also one of the main weaknesses of the manuscript as it is suggested in the introduction and the abstract that this is the contribution of the current work. Additionally, the discussion section does not concretely specify the contribution and the outcome of the reproducibility of the benchmark. For example, it specifies that the extended explanations would improve the diversity of the audience for potential users of the benchmark, but it does not specify how it does so, and how that is represented in the metrics used in the original benchmark or the current manuscript. In other words, are the metrics different from the original manuscript? Should different users use different metrics? Is there some adapted metric for specific users? Did the current manuscript overcome the limitations mentioned in the original benchmark and was able to produce more simulation games automatically? And so on…

---

> ### Author Response · Authors · 2025-04-08
>
> We appreciate the thoughtful feedback and have revised the manuscript accordingly. For clarity, we address each point separately:
>
> 1. **Addressing the Leakage Bug**
> The reviewer notes that our postprocessing to filter leakage (or assist smaller models) seems like an extra engineering step rather than reflecting the models’ intrinsic behavior. This may appear to contradict our claim that the leakage metrics stem from code limitations rather than the model itself. We note that the term “leakage” in the original paper conflates two failure modes:
> (a) *Theory-of-Mind (ToM) Failure*—where a model inadvertently discloses private information, and
> (b) *Structural Formatting Failure*—where the model fails to follow the prescribed output structure.
> In the revised manuscript, we explicitly separate these issues, clarifying that our postprocessing targets only the structural formatting failure—a behavior that is intrinsic but must be evaluated separately from higher-level reasoning such as ToM. This distinction permits a clearer benchmark of a model’s higher-order capacities independently from its template-following ability. We have refined the text to ensure clarity.
>
> 2. **The Adjustment Claim and Context Bias**
> The reviewer points out that while our manuscript highlights context bias in generating new games, results in Table 3 show that some models (e.g., Qwen2.5-72B) outperform others (GPT4o-mini, Mistral-small), seemingly contradicting the original negotiation claim. Our main claim is that benchmark consistency varies across models. Although Qwen outperforms GPT-4o mini overall, Table 3 shows that Qwen underperforms Mistral on game2, suggesting that intrinsic game features affect model performance differently. We propose that these features contribute to context bias in generation, indicating that certain game elements may favor one model over another. We clarify these nuances and discuss their implications for future benchmarking.
>
> 3. **Clarity on Future Work and Reproducibility Contributions**
> The reviewer criticizes the discussion of future work for lacking clear specifications on our contributions and reproducibility outcomes, noting uncertainty over whether our new metrics differ from the original ones and if different users should adopt different metrics. We have rephrased this section to explicitly outline our contributions. We show that certain bugs (e.g., acceptance condition inconsistencies and threshold relaxations) in the original code lead to erroneous results. Correcting these issues makes the benchmark more accurate for models that struggle with formatting adherence. Our additional metrics (including sparsity and Intersection-over-Union) are used solely to characterize the provided games and evaluate their diversity. Our goal is to perform a reproducibility check that validates the original benchmark while highlighting its limitations. We clearly state that our work does not advocate for an entirely new set of evaluation metrics but calls for a more nuanced interpretation of model performance and game understanding.
>
> 4. **Revising the Abstract**
> The reviewer finds the abstract ambiguous—particularly in clarifying model comparisons relative to the original benchmark—and notes that terms like “leakage detection” and “ablation transferability” are vague. We have revised the abstract to be more concrete and aligned with our discussion and results.
>
> 5. **Aligning the Discussion with the Abstract and Introduction**
> The reviewer observes a mismatch between the discussion (which focuses on minor fixes) and the abstract/introduction (which emphasize critical issues), and is concerned about mixing design issues with coding mistakes. We have restructured the discussion to mirror the issues raised in the abstract and introduction. We now clearly separate design issues from coding mistakes and detail the critical issues and their impact on reproducibility, providing a coherent narrative that reinforces our findings.
>
> 6. **Clarifying the Contributions Section**
> The contributions section did not accurately reflect the experimental outcomes and discussions. We have revised it to align more closely with the experimental results and the revised discussion.
>
> We trust these revisions address the reviewer’s concerns. We have aimed to offer a more precise and transparent account of our methodology, contributions, and the implications of our reproducibility study, and we look forward to any further feedback.

---

### Review · Reviewer_eqCD · 2025-03-26

**Summary Of Contributions:**

The paper is a reproducibility report of the work of Abdelnabi et al.(https://proceedings.neurips.cc/paper_files/paper/2024/file/984dd3db213db2d1454a163b65b84d08-Paper-Datasets_and_Benchmarks_Track.pdf) which appeared last year at NeurIPS datasets and benchmarks track which was a benchmark for negotiation between multi-agent LLMs. The paper reproduces some of the results in the paper. It also extends the evaluation to other models. It discusses additional metrics. It fixes minor bugs in the calculations of the final deal of the original code.

**Audience:**

Yes

**Claims And Evidence:**

No

**Requested Changes:**

- I believe the conclusion of the paper and the main message in the introduction should be more consistent. At the moment, the paper exaggerates the contribution by making unsupported claims and refuting claims were not introduced in the original work.

- The paper needs to discuss the limitations of their approach in calculating the score leakage.

- It would be of great value to elaborate on the many points raised in the discussion on how to extend the original work. For example, how to systematically (and objectively) assess the diversity of games, not only based on overall scores but pairwise dynamics between players and semantic role. Another point how to change thresholds automatically. And how to extend the work to study other attacks and safety aspect. By that I believe the paper would serve advance the field. At the moment, the paper raises many points, some of them were raised by the original work, while not introducing an attempt for improvement.

**Strengths And Weaknesses:**

Strengths:
========
- The paper introduces new metrics that can give new insights based on the notion of social welfare (although the paper does not elaborate on how to interpret them to evaluate the agents or the difficulty of games)

- The paper identifies improvements to fix problems related to models not following correct formats (for example, making sure the agreement is properly calculated based on the final deal and that the final deal is correctly formatted and not carrying over a previously parsed deal as the final one).

- It also improves the evaluation of whether the scores were leaked by making sure to evaluate this on the correctly formatted public answers while reporting failures in formatting (although I disagree that the paper in the correct form evaluates leakage correctly and it significantly oversimplified the leakage to formatting, rather than more broadly as how models preserve theory-of-mind and contextual integrity principles, details on that in the next section).

- The paper extends the evaluation to many other models, especially new ones such as DeepSeek. This is helpful to track the performance of reasoning models.

- I found the discussion of the paper valuable on breaking down negotiation to specific sub behaviors.

- The paper points a minor issue in the writing of the original paper (must exceed) while the code is greater than or equal. However, the code is consistent with the original game (Susskind et al.)

------------------------------------------------------------------------------------------

Weakness:
========

While I think the paper introduces valuable experiments and discussions, in many situations, I find the paper to overclaim the contributions made, oversimplify the metrics and provide them as better alternatives, and make wrong claims about the original paper. For a reproducibility report, the paper assumes claims that were not stated in the original work and then refutes them. However, they were not really claimed. The paper also extends the discussion with many limitations, some of them are already raised and acknowledged in the original paper, as criticism of the original work. Discussions about extending evaluating the negotiation are valuable, but framing them within the context of reproducibility is misleading to the reader. The conclusion of the paper itself does not support the claims made in the introduction. The conclusion mentions "Additionally, while the various minor bugs in the original implementations do not invalidate the results of the original paper, they potentially pose problems to users evaluating weaker models. By fixing them, we improve the diversity of the audience of potential users of this benchmark.", but this is not the main message the rest of the paper implies.

Therefore, I think there is a room for overall improvement of the paper with clarification, rewriting, and further discussions. I elaborate on this and other points below.

## Minor:

- The abstract is not clear, the use of the word "fairness" implies it is related to biases and fairness of decision making, while the rest of the paper show it is actually about trying to unify the evaluation and comparison between different models.

-  Related to the previous point, the introduction mentions "leakage risks". The paper didn't show how this benchmark may have leakages risks and mentioning it in the introduction biases the reader into believing the paper is actually about leakage. These two points are minor but it is related to it is not clear from the abstract and introduction to know what the scope of the paper actually is.

## Motivation:

- In the introduction, the paper implies that the "non-viability of the benchmark on many weaker models" weakens the ability of the benchmark to evaluate models. I disagree with the premise of the paper that all benchmarks need to be tailored to ALL models. I believe we need complex benchmarks for specifically frontier models and that is where the gap is. There exists previous simples benchmarks for negotiation (Davidson et al. https://arxiv.org/abs/2401.04536). That being said, Abdelnabi et al.'s benchmark (the games itself) can still be used for evaluating smaller models. There are two pillars that the paper are confusing: the games themselves, and the metrics. While a metric such as the agreement rate is discrete and might show all smaller models are equally bad, one can propose other metrics using the same games (such as following formats, not leaking information, performing correct calculations, mapping issues to correct scores, making correct inferences about other players' preferences, etc.). Future work could use the benchmark (the games) and augment the metrics in the original paper for more evaluation.

## Misleading contribution claims

- " we identify a more diverse set of negotiation settings, including discussions on military spending budgets for the upcoming year and planning a conference, where aspects like floor allocation and catering must be negotiated." The original work followed this prompt setting with manual curation of games and rewriting, as mentioned in the original paper. I don't believe the paper makes a contribution in having end-to-end negotiation games, just the seed.

## Oversimplification of metrics

- In 3.1.5, "addressing leakages", as mentioned above, performing the analysis on strictly correctly phrased public answer is a good and valid addition (while still reporting the rate of failures because this an important property of evaluating the model). However, the paper seems to consider leakage only when "scratchpad" or "plan" tags appear. I disagree with this approach. In fact, using GPT4 evaluation is the better way to do this. In many cases, models may use the proper format but still mentions the score ("as it provides a mutually beneficial agreement that meets the minimum score requirement of 50." -- see Llama2 70b output https://github.com/S-Abdelnabi/LLM-Deliberation/blob/main/logs/llama2_70b_base_all_coop.zip, or even mentioning "getting closer to my minimum score requirement", ). This should be detected because 1) it is a failure in following instruction, 2) it is not aligned with how humans negotiate by not revealing their priorities and plans. Also, in many cases, it indicates a theory-of-mind limitation, see Appendix I, as advocating for a deal with a higher score for a party in the public answer may indicate the models are not correctly processing that other parties have different priorities (I endorse Deal 12 ( A1, B1, C3, D3, E4 ) as my final proposal. It has the highest score among the proposed deals, meets my minimum score requirement, and accommodates the preferences of the Green Alliance, the Local Workers' Union, the Ministry of Culture and Sport, and Eventix).

## Evaluation choices that are both valid and are not weaknesses of the original work

- The paper excludes the setting of relaxing the threshold of p1 if leading to all party agreement, this is an experimental choice which is valid. However, the original paper follows Susskind et al. which has the same condition. In Abdelnabi et al.'s setup, this information was given to p1. In cases where GPT-4 is cooperative, it may perform correct calculations and reach a correct conclusion that the scores are less than the threshold. However, it can add the 10 point since it is attempting to have an all-group agreement. For this reason, it can make sense to consider this for checking agreement to accommodate for this. Removing this for checking agreement is valid, but should be removed as well from p1's prompt.

- For baselines (in 3.2.1), as far as I know, there are no clear canonical baselines for the game. A random order is valid. However, the "priority" of issues is not undefined, this can be based on the maximum score that can be achieved by this issue. -	In Table 4, I don’t understand on what factors this baseline provides more accessibility and explainability. The difference between this baseline and the one in the original paper is using random ordering instead of sorting. I agree however with the paragraph after about that rule-based heuristics do not capture the complexity of human negotiations.

## Incorrect claims about the original work

- For 3.2.2 Adjustments claim, Abdelnabi et al. didn't claim that the number of the set of feasible deals size is the main diversity factor between the different curated games. In fact, the paper mentioned explicitly that scores were adjusted to have very similar numbers to the base game ("we assigned numerical scores to reach a comparable ratio of feasible deals compared to the base game", section 3). In my opinion, this is valid, as since the random chance ratio of finding an agreement is the same, one can make further analysis of why certain games may seem to be easier than others. Abdelnabi et al. didn't also claim that sparsity can be easily tuneable, section (4.6 Tuning the Game Difficulty) in the original paper only mentions adjusting minimum scores. However, they claimed that the pre-defined and created games have (fixed, curated) diversity in terms of sparsity. They claim that "game 1" and "game 2" are less sparse compared to the "base" game and "game 3" (this is actually reproduced by table 5) in this paper. The original claim of creating diversity by changing thresholds seems to be reproduced with this paper (Fig. 2)

- I don’t disagree that different prompts may work better with other models. As far as I can tell, Abdelnabi et al.’s paper didn’t claim that best prompt structure would generalize. However, the paper didn’t really provide a bigger picture of trends and didn’t provide enough context of how this would change any major conclusions. It is not really clear from table 2 that these yellow configurations are better than the red one (for example, they have 0% 6-way agreement for Qwen-72B). Even under the same model, given that experiments are run for 20 times for each model which is understandable due to the size of each game, these small deviations are expected. If the intention is to show very thorough comparison of models rather than showing general trends, yes, the ablation should be done for each model. Additionally, experiments should be run for more rounds.

- “The results clearly demonstrate that the negotiation scenarios generated with the original prompt are not diverse, and simply reusing the prompt does not lead to greater scenario variety”. I believe this claim is not supported by defining what “diversity” is. Sure, having more scenarios other than constructing a project is an additional source of diversity, however, this does not necessarily mean than within the same domain, one can have other sources of diversity (in the semantics and relationships, etc.). The paper can claim they provide more diversity, but it shouldn’t overclaim without evidence the lack of diversity of the original work. Of course, if one needs to create a significantly large sample of games, more systematic analysis should be done to say how diverse/redundant they are. But, Abdelnabi et al. didn’t claim they provide a mechanism to create new games end-to-end. The games proposed are fixed and manually curated. The discussion of how to extend that is important, but not within the context of reproducibility. The original paper already acknowledged the limitation of scaling and changing scores and it does not claim that creating diverse games or changing scores automatically were done.

## Other points
- The results in table 3 are interesting, but the paper does not provide further analysis and observations. For example, about Mistral’s bad performance on game3. That may provide further insights.

- “game difficulty is not an inherent property of the game itself but is instead directly influenced by the model being evaluated” one can identify several properties of the game itself (for example, sparsity being one of them).

- The paper mentions "Additionally, it complicates the application of established formalisms, such as Nash
Equilibrium analysis, which is commonly used in other benchmark studies (Brookins and DeBacker, 2023).
The absence of such formal criteria further hinders the objective assessment of benchmark results." That can be true, I don't disagree. However, the paper does not take into consideration that these games used in the original work were not invented by the authors (as it is clearly mentioned in the paper). It is a very old exercise for teaching negotiation that is still used now. Beyond rational players, this can have many potential opportunities to study parallels between humans and LLMs and also the communication aspects inspired by behavioral game theory.

---

> ### Author Response · Authors · 2025-04-08
>
> We thank you for your detailed review. Your feedback has helped us clarify and improve the manuscript. Below, we detail our responses and the revisions:
>
> **1. Claims and Attribution:**
> We revised the text to avoid implying that Abdelnabi et al. made claims regarding ablation setting generalization or automatic game diversity. Instead, we now present these points as independent extensions rather than critiques of the original paper. Specifically, we removed statements suggesting that Abdelnabi et al. claimed generalization of ablation settings across models or inherent diversity through automatic game generation.
>
> **2. Reproducibility vs. Extensions:**
> Our introduction and conclusion now clearly separate our reproducibility goals from novel contributions. The manuscript explicitly distinguishes between replicating the original methodology and introducing further analyses.
>
> **3. Abstract and Introduction Clarity:**
> We adjusted the wording in both sections to clarify that “fairness” refers solely to equitable model comparisons, not decision-making fairness, and that “leakage risk” is limited to format-based leakages.
>
> **4. Leakage Evaluation Limitations:**
> We thank the reviewer for acknowledging our separation of structural leakage from overall leakage as an added contribution. It is true that with our method, more subtle leakages like that described in the appendix of the original paper are not detected and an additional pass by a powerful model like GPT-4 would aid in catching those. However, we would like to emphasize that the total number of structural leakages we removed composed the vast majority of the detected leakages in the original work. A more comprehensive ToM assessment applied after the adjustment would then produce more illuminating and valid results. Our position though is that, as part of a reproducibility study, it was enough to demonstrate this key confounding factor and further analysis may be left to future work. We have adjusted our manuscript accordingly to reflect this fact.
>
>
> **5. Evaluation Metrics and Diversity:**
> Regarding diversity of game settings, we agree we should not overclaim limitations of the original work. We would like to clarify our position, our examination highlights the potential bias towards construction-related scenarios. We are aware that Abdelnabi et al. did not claim to offer automatic or end-to-end diversity in game creation, and we do not aim to tackle that. We corrected our manuscript to acknowledge that  the set of feasible deals was similar by design.
> As to the diversity of game statistics, our primary objective was to formally evaluate the diversity of the negotiation games. To accomplish this, we proposed several statistical measures intended to capture potential differences across games. We found these statistics to vary within a certain range, which provides partial evidence of diversity. However, the extent to which this range can be considered sufficiently broad is open to interpretation.
>
> **6. Ablation Study and Generalization:**
> Although Abdelnabi et al. do not explicitly claim that their ablation configurations generalize, their design—using the best configuration for GPT-4 across models—implies this assumption. We tested this by examining performance variances across setups and updated Table 2 to clarify that our objective was not to find alternative optimal configurations, but to assess variability in comparative performance.
>
> **7. Difficulty and Model Dependence:**
> We rephrased our discussion on game difficulty to clarify that, while inherent game properties (e.g., sparsity) exist, the difficulty—as measured by performance rankings—varies with the model used in different negotiation scenarios.
>
> **8. Threshold Relaxation:**
> We clarified our concern regarding the threshold relaxation issue. The inconsistency between the textual description and its implementation in the original paper suggests that this parameter should be removed from the code, as it lacks proper justification. It is never mentioned in the original paper except for one caption in the appendix, nor is it mentioned to the agents in the prompts.
>
> **Additional Revisions:**
> - **Baseline Clarification:** We now state that our baseline experiment was designed solely to provide a fully reproducible baseline.
> - **Future Work:** We expanded our discussion on future directions.
> - **Human Benchmark Usage and Nash Equilibrium:** We added a note recognizing the relevance of human benchmarks and behavioral game theory, suggesting these as promising areas for future study.
> - **Conclusion Consistency:** The conclusion now mirrors the contributions outlined in the introduction, emphasizing methodological improvements, reproducibility verification, and the introduction of new analytical metrics rather than an overall critique of the original benchmark.
>
> We appreciate your constructive comments and believe these revisions enhance the manuscript’s clarity, accuracy, and depth.

---

> > ### Comment · Reviewer_eqCD · 2025-05-08
> >
> > Thank you for your detailed response and I apologize for the late reply.
> >
> > The revision significantly improves the paper and I appreciate the changes.
> >
> > I now agree with the leakage metrics.
> >
> > I have the following comments:
> >
> > ## Threshold relaxation:
> > - It is implemented in the code (please see here: https://github.com/S-Abdelnabi/LLM-Deliberation/blob/main/initial_prompts.py)
> > - It is part of the original game (https://www.pon.harvard.edu/shop/harborco/)
> > - It is again a design choice whether to include this relaxation or not. It is not a limitation of the original work or the current work if it is included or not included. But if it is included in the prompt, the evaluation needs to take this in consideration. As far as I see, Abdelnabi et al. include it in the prompt and evaluation.
> >
> > ## Adjustment and diversity:
> > - I believe these are two separate issues. I don't disagree with the paper on the diversity evaluation. But the original work claimed that games are "easily adjustable". The paper mentions: "This framework provides a collection of games that are diverse and easily adjustable. This will be referred to as the adjustments claim. We show that this claim is mostly untrue, with caveats." The paper didn't show that games are not easily adjustable (by changing the thresholds of parties). In fact, the paper agrees with the original work regarding adjustment, quoting the paper: "The results for Experiment 7 are shown in Figure 2, where we examine how easily adjustable games are using the modifiers provided by original authors. The results are consistent with the findings of the original paper: in Figure 2(a) performance tends to increase consistently with a decrease in the minimum threshold per player. Similarly, we find that more agreements are reached with a lower number of players, see Figure
> > 2(b)."
> > - As a result, I think the paper should clearly separate these two issues to be consistent with its own findings.

---

> ### Author Response · Authors · 2025-05-12
>
> Thank you again for your constructive feedback and for your patience as we undertake this further revision. Below, we address each of your remaining points in turn:
>
> 1. Threshold relaxation
> We have revisited Susskind’s original paper [1] and the official game rules [2] available on the Harvard Law School website. We were not able to find any mentions of this relaxation, could the reviewer please provide the exact source or section where threshold relaxation is discussed? In the initial_prompts.py file, the accompanying comment states:
> “To protect yourself from potential future lawsuits, you want to achieve unanimity; if you and all other {self.num_agents-1} parties agree, you will get a bonus of 10 points.”
> However, the existing code grants Player 1 a 10-point threshold relaxation regardless of whether unanimity is achieved. To resolve this inconsistency, we have now modified the implementation so that Player 1 receives the 10-point bonus only when all other agents agree. We have rerun our experiments under this corrected rule set and updated the results accordingly. Should this reflect the intended gameplay, we would be happy to include the revised tables and figures in the next draft.
>
>
> 2. Adjustments versus diversity
> We agree that the distinction between “adjustability” and “diversity” was not sufficiently clear. We have revised that section to separate these concepts more explicitly and evaluated independently. We believe this clarification improves the readability and conceptual rigour of our presentation.
>
> We trust these changes address your concerns and enhance the manuscript’s clarity. Thank you once again for your insightful suggestions; they have been invaluable in strengthening our work.
>
> [1] https://direct.mit.edu/ngtn/article/1/3/205/123101/Scorable-Games-A-Better-Way-to-Teach-Negotiation
>
> [2] https://www.pon.harvard.edu/shop/harborco/

---

### Decision · Action_Editor_eHyd · 2025-05-19

**Recommendation:** Accept with minor revision

**Comment:**

This submission is to the Reproducibility track, where the emphasis is on insights and reproducibility rather than novelty. The reviewers acknowledge that the authors have addressed most concerns in the revision. While some ambiguity remains regarding the novelty and interpretation of certain design choices, the paper appears to meet the acceptance criteria for a reproducibility submission. Most remaining issues relate to writing and clarity, which are fixable. Overall, the paper makes a meaningful contribution, and I recommend acceptance upon minor revision.

**Audience:**

Given the value of the reproducibility effort and the insights delivered, I believe this paper will be of interest to researchers engaged in related areas.

**Claims And Evidence:**

The paper reproduces the results of “Cooperation, Competition, and Maliciousness: LLM-Stakeholders Interactive Negotiation”, which appeared in the NeurIPS 2024 Datasets and Benchmarks track as a benchmark for negotiation between multi-agent LLMs. In addition to reproducing key results, the authors update the original benchmark with recently released models, fix minor issues in the original code, and extend the evaluation to include additional metrics and insights. The paper also offers expanded discussion on the benchmark’s design, the negotiation dynamics involved, and outlines promising directions for future metric development.

---

> ### Author Response · Authors · 2025-06-02
> **Final clarifications**
>
> Thank you very much for your time and for recommending our submission for acceptance with minor revision. We appreciate the Action Editor’s and reviewers’ careful reading and feedback.
>
> Before we finalize the camera-ready version, could you please clarify which specific parts of the manuscript still need adjustment? In our revised draft we believe we have already addressed each of the reviewers’ comments—some sections were rewritten, and in a few cases we provided explanations to justify our design choices—so it is not immediately clear to us what remains to be improved. You mention that most remaining issues relate to writing and clarity. If there are particular sentences, figures, or arguments that you feel are still unclear or require rewording, we would be grateful for any examples or pointers.
>
> Having these details will allow us to focus our edits precisely and ensure that the camera-ready version fully addresses all outstanding concerns. Thank you again for your guidance, and please let us know if there is anything else we can provide.
>
> Best regards

---

> > ### Comment · Action_Editor_eHyd · 2025-06-03
> >
> > I believe you've addressed most of the technical feedback during the rebuttal, which is great. However, the writing and formatting can still be further improved for the final version.
> >
> > For example, I noticed a lot of unnecessary white space in the appendix (e.g., page 19), particularly between different sections. This happens when \newpage or \clearpage commands are used excessively.